# Learning Multimodal Representations from Partially Paired, Small-Scale Data

## Abstract

Modeling multimodal data from partially paired samples is critical for advancing domains like biomedicine, where vast unimodal datasets and foundation models exist but paired data remains scarce. Existing fusion methods rely on large-scale paired datasets, limiting their use in scenarios with incomplete pairing. We introduce CAMEO, an adversarial learning-based modality fusion framework that integrates modalities from small, partially paired datasets. Combining any pre-trained unimodal encoder with a cross-modal latent alignment mechanism, CAMEO learns shared representations requiring only minimal paired samples. Evaluated on computational pathology tasks such as niche classification and cell type composition prediction, CAMEO achieves superior data efficiency, outperforming contrastive approaches like CLIP in low paired-data regimes, and highlighting the benefits of adversarial alignment when paired annotations are scarce. To facilitate further research, we additionally release a fully annotated HuggingFace dataset comprising three organs and paired image and gene expression modalities. By extending fusion methods to address limited pairing and small-scale datasets, we provide a framework that advances multimodal learning and broadens its applicability to real-world biomedical problems.

## 1 Introduction

Recent advances in spatial transcriptomics and histopathology (HP) imaging have advanced our understanding of tissue organization by providing complementary views of cellular states (Larsson et al., 2021; Palla et al., 2022; Fischer et al., 2023). While spatial transcriptomics captures gene expression with spatial information, HP imaging reveals tissue morphology and cellular organization at high resolution. The integration of these modalities promises deeper insights into tissue architecture and cellular interactions.

Foundation models have emerged as powerful tools for analyzing biomedical modalities (Szałata et al., 2024). These models leverage self-supervised learning on large, unimodal datasets (Richter et al., 2024). Models like scGPT (Cui et al., 2024a) and Geneformer (Cui et al., 2024b) learn gene-gene dependencies from gene expression data (i.e., quantitative transcriptomic profiles) through masked language modeling. This type of model is applicable to various cell- and gene-level downstream tasks, including cell annotation and gene function prediction. Vision models like UNI (Chen et al., 2024) and CTransPath (Wang et al., 2022) learn morphological patterns from H&E-stained whole-slide images using a contrastive objective. However, these approaches do not utilize the potentially complementary information present in multiple modalities. While multimodal models have been successfully applied in domains such as language and vision (Radford et al., 2021), they typically require large-scale paired datasets, limiting their applicability to domains where paired data is scarce.

We introduce CAMEO (Fig. 1), an adversarial modality fusion framework, as an alternative to contrastive and naive concatenation approaches. In biomedical settings where paired samples are scarce, CAMEO's adversarial mechanism provides robust performance with minimal paired data requirements. Our cross-tissue analysis reveals the complementary strengths of different fusion strategies, with adversarial alignment offering superior robustness when paired data is limited, while also highlighting the importance of systematically evaluating encoder combinations regardless of fusion approach.

Figure 1: CAMEO model. Top left: Cells in a tissue are represented by various modalities, including histology (vision), gene expression vectors (functional biology), and further metadata such as expert annotations that we use for downstream tasks. Top right: The model is first pre-trained on widely available, unpaired, unimodal data from vision (top) and gene expression (bottom) repositories, and then fine-tuned on small-scale paired datasets. Bottom: Unimodal image and gene expression encoders are trained with self-supervised learning to map images to $z_I$ and gene expression vectors to $z_G$. CAMEO then projects these generated representations into a shared multimodal latent space via $E_{sh}$. This allows prediction of the gene expression representation from the image representation and vice versa.

Our comparative evaluation offers practical insights for choosing multimodal integration strategies in computational pathology. In summary, we present the following contributions:

- We introduce CAMEO, an adversarial modality fusion framework that aligns pre-trained unimodal encoders in a shared latent space, effectively leveraging small, partially paired datasets.

- We release an aligned, expert-annotated HuggingFace dataset combining image and gene expression modalities across three tissue types.

- We demonstrate that multimodal approaches generally outperform unimodal baselines on niche classification tasks and that the ideal combination of encoders is dictated by the dataset.

- We reveal that a unimodal encoder's standalone accuracy is a poor predictor of its ultimate benefit in a multimodal setting. This highlights the need to systematically evaluate encoder combinations rather than assume that the best single-modality models will automatically yield the best fused models.

- We show that CAMEO exhibits superior robustness to reduced pairing percentages compared to CLIP and concatenation baselines, particularly valuable in scenarios with limited paired data and class distribution shifts.

## 2 RELATED WORKS

### 2.1 MULTIMODAL TRAINING ON PAIRED MODALITIES

Multimodal representation learning algorithms typically rely on paired data to align embeddings across modalities. For instance, CLIP (Radford et al., 2021) is trained on a massive corpus of 400 million image-text pairs, showcasing impressive performance in zero-shot classification tasks. Similarly, ImageBind (Girdhar et al., 2023) aligns multiple modalities under the condition that samples are represented by at least one common modality, which can be restrictive when such overlaps are scarce. BLIP-2 (Li et al., 2023) aligns visual features with a pre-trained language model through a Querying Transformer, leveraging paired image-text data for training. Similarly, LLaVa (Liu et al., 2023) fine-tunes pre-trained models on paired image-text data to enable complex reasoning. CoCa (Yu et al., 2022), on the other hand, is trained from scratch, utilizing paired data to support multimodal tasks like captioning and cross-modal alignment.

## 2.2 MULTIMODAL APPROACHES IN PATHOLOGY

In pathology, multimodal methods have leveraged the integration of imaging and textual data to improve performance in diagnostic and prognostic tasks. These approaches generally fall into two regimes: patch-level and slide-level. Patch-level vision-language methods (Huang et al., 2023; Lu et al., 2024a;b; Zheng et al., 2024; Ikezogwo et al., 2025) typically adopt CLIP-style contrastive pre-training. In contrast, slide-level methods such as TITAN (Ding et al., 2024), PathAlign (Ahmed et al., 2024), and PRISM (Shaikovski et al., 2024) leverage multiple instance learning (MIL) algorithms on paired whole-slide image and text data to perform, for example, pathology report generation and cross-modal retrieval of histology slides and clinical annotations.

In addition to image-text data, image-expression data has also been explored in multimodal pathology approaches. Tangle (Jaume et al., 2024) combines histology images with paired Bulk RNA sequencing data for few-shot slide classification and retrieval, showing improved performance over unimodal baselines on slide-level tasks. This highlights the benefit of integrating high-dimensional sequencing data with imaging data. RNA-CDM (Carrillo-Perez et al., 2024), HE2RNA (Schmauch et al., 2020), and BLEEP (Xie et al., 2023) explore the translation between imaging and sequencing modalities, generating one modality from instances of the other. Yang et al. (2021) uniquely enable this translation with unpaired data. Survival prediction tasks have also benefited significantly from multimodal approaches, as exemplified by MOTCat (Xu & Chen, 2023), CMTA (Zhou & Chen, 2023), GPDBN (Wang et al., 2021), and HFBSurv (Li et al., 2022). These methods integrate paired imaging and sequencing data to build predictive models that outperform unimodal counterparts in terms of clinical outcome prediction. More recent methods, such as MUSK (Xiang et al., 2025), relax the need for fully paired data by pre-training on large-scale unpaired datasets. While achieving strong performance in precision oncology tasks, MUSK's dependence on large corpora for robust cross-modal embeddings may limit its generalizability when only a small amount of data is available.

## 2.3 MODAL ALIGNMENT AND INTEGRATION TECHNIQUES

Recent alignment work has sought to integrate unpaired multimodal samples by incorporating prior knowledge or causal inference techniques. GLUE (Cao & Gao, 2022) employs a knowledge graph to guide the integration of unpaired multi-omics data, offering an interpretable framework for combining disparate modalities. However, its generalizability is constrained when constructing an appropriate knowledge graph is impractical or infeasible. Xi et al. (2024) estimate a propensity score in each modality, capturing shared information between latent states and perturbations. Though this method enables matching unpaired samples across modalities, it still requires robust data curation. Another line of research has focused on using optimal transport to align unpaired omics datasets. Methods such as SCOT (Demetci et al., 2022) and GROT (Wang et al., 2024) leverage the shared gene feature space to achieve cross-modal alignment, but gene expression and histology images lack a common coordinate system, making it difficult to define a biologically meaningful transport cost between them. Nakada et al. (2023) extend the CLIP loss to incorporate unpaired data, enabling alignment of paired images sampled from MNIST and Fashion-MNIST. This synthetic dataset might oversimplify multimodal relationships, limiting the generalizability of the findings to more complex natural data distributions.

Our approach builds on this growing body of multimodal alignment research. However, we focus on real-world applications in clinical pathology where data is limited and paired samples are scarce due to the expense of obtaining them. We specifically target patch-level tasks rather than slide-level analyses. Since the latter rely on MIL algorithms, they are unsuitable for patch-level tasks. Unlike previous patch-level methods trained on paired vision-language datasets, our approach leverages both paired and unpaired vision-transcriptomics data. By using pre-trained unimodal encoders and a cross-modal alignment mechanism, we enable efficient adaptation to downstream tasks with minimal paired labeled examples. This capability is particularly relevant in medical settings, where collecting large amounts of paired multimodal data can be both costly and time-consuming. Beyond demonstrating effectiveness on downstream tasks, we conduct systematic ablation studies across a range of gene expression and image encoder combinations. We find that multimodal models typically achieve better performance than unimodal baselines, though results depend on the combination of encoders used. Importantly, the effectiveness of unimodal encoders alone does not consistently forecast their success within multimodal models, emphasizing the need for thorough evaluation when choosing encoders for multimodal approaches.

## 3 METHOD

### 3.1 OVERVIEW

CAMEO is modular by design and compatible with any image or gene expression encoder. It follows a three-stage process: (1) initial data embedding with modality-specific encoders, (2) unsupervised representation alignment via cross-modal generation, and (3) supervised evaluation via linear probing for downstream tasks. By using pre-computed latent embeddings instead of raw data, we reduce computational complexity and memory requirements. CAMEO's design allows us to incorporate powerful pretrained encoders, such as foundation models trained on large-scale histology or single-cell data, which can improve robustness and generalizability. Our approach addresses several key challenges in multimodal self-supervised learning, including limited data availability, which makes assembling large datasets costly and technically difficult; scarcity of paired data, which constrains models that require fully paired examples such as CLIP; and batch effects, arising from systematic variations introduced by different experimental protocols and equipment.

Next, we provide a detailed description of the individual components of the model: 1) the vision encoder, 2) the gene expression encoder, 3) the pretraining process using multimodal fusion, and (4) the linear probing stage for classification and regression.

### 3.2 VISION ENCODERS

Let $x^I \in \mathbb{R}^{H \times W \times 3}$ denote an image patch extracted from a tissue slide, where $H$ and $W$ are the height and width of the patch, and the 3 channels correspond to RGB color intensities. Each image shows the morphological structure of the cells present at an average magnification of 20x and a resolution of 0.5 microns per pixel (MPP). Modern foundation models for pathology often adopt Vision Transformer (ViT) (Dosovitskiy et al., 2021) architectures pre-trained on large cohorts of histology slides. To evaluate the role of the vision encoder in multimodal fusion, we conduct an ablation using five pretrained models (four ViT-based encoders and one convolutional encoder). These encoders are used in a plug-and-play fashion within the CAMEO framework. Further details on the specific encoders used are provided in Section 4.3.

### 3.3 GENE EXPRESSION ENCODERS

Let $X^G \in \mathbb{R}^{n \times p}$ denote the gene expression matrix, where $n$ is the number of cells or spots and $p$ is the number of measured genes. Each input $x^G \in \mathbb{R}^p$ is a high-dimensional vector representing the expression profile of a single cell or spot. Gene expression encoders transform these vectors into compact latent representations that capture biological signals relevant for downstream tasks.

To systematically assess how different gene encoding strategies impact performance, we conduct an ablation over five gene expression encoders. This includes four existing models from the literature, as well as a Graph Attention Network (GAT) that we train ourselves (Richter et al., 2023). The GAT encoder follows the formulation of Graph Attention Networks (Velickovic et al., 2017), learning node embeddings by aggregating information from spatial neighbors using learned attention weights. We pretrain the GAT using a self-supervised masked modeling objective: random subsets of each expression vector are masked and reconstructed from unmasked entries using the neighborhood structure. All encoders are used in a plug-and-play manner, and details on their usage and evaluation setup are provided in Section 4.3.

### 3.4 MULTIMODAL FUSION

In this work, we propose a general adversarial framework for learning cross-modal representations.

#### 3.4.1 MODEL ARCHITECTURE

The adversarial fusion module consists of two cross-modal generators ($\mathcal{G}$) and their corresponding discriminators ($\mathcal{D}$):

$$\mathcal{G}_{G \to I} : \mathcal{X}_G \to \mathcal{X}_I \qquad\qquad \mathcal{D}_I : \mathcal{X}_I \to \mathbb{R}$$
$$\mathcal{G}_{I \to G} : \mathcal{X}_I \to \mathcal{X}_G \qquad\qquad \mathcal{D}_G : \mathcal{X}_G \to \mathbb{R}$$

Here, $\mathcal{X}_I \subset \mathbb{R}^{d_I}$ and $\mathcal{X}_G \subset \mathbb{R}^{d_G}$ denote the embedding spaces of images and gene expression, respectively. The inputs to the generators are $z_I \in \mathbb{R}^{d_I}$ and $z_G \in \mathbb{R}^{d_G}$, which are obtained from modality-specific unimodal encoders:

$$z_I = F_I(x^I), \quad z_G = F_G(x^G),$$

where $F_I : x^I \to \mathbb{R}^{d_I}$ is the image encoder and $F_G : x^G \to \mathbb{R}^{d_G}$ is the gene expression encoder. Each generator employs a shared bottleneck architecture:

$$\mathcal{G}_{G \to I}(z_G) = E_{sh}(E_G(z_G))$$
$$\mathcal{G}_{I \to G}(z_I) = E_{sh}(E_I(z_I))$$

where $E_G$ and $E_I$ are projection layers that adapt expression and image embeddings, respectively, before passing them into the shared layers $E_{sh}$.

### 3.4.2 Training Objectives

**Stage 1: Representation Learning** In the first stage, we focus on learning shared representations from partially paired image and gene expression data. Given paired samples $\{(x_i^I, x_i^G)\}_{i=1}^{n_p}$ and unpaired samples $\{x_j^I\}_{j=1}^{n_u^I}$, $\{x_k^G\}_{k=1}^{n_u^G}$, our goal is to learn mappings between modalities that preserve semantic relationships.

The objective of CAMEO consists of three components:

1. **Adversarial Loss**: For each modality, a discriminator learns to distinguish real from generated embeddings (Goodfellow et al., 2014):

$$\mathcal{L}_{WGAN} = \mathbb{E}_{z_I \sim p_I}[\mathcal{D}_I(z_I)] - \mathbb{E}_{z_G \sim p_G}[\mathcal{D}_I(\mathcal{G}_{G \to I}(z_G))]$$
$$+ \mathbb{E}_{z_G \sim p_G}[\mathcal{D}_G(z_G)] - \mathbb{E}_{z_I \sim p_I}[\mathcal{D}_G(\mathcal{G}_{I \to G}(z_I))]$$

2. **Cycle Consistency Loss**: To ensure semantic preservation during cross-modal translation (Zhu et al., 2017):

$$\mathcal{L}_{cycle} = \mathbb{E}_{z_I \sim p_I}[\|\mathcal{G}_{G \to I}(\mathcal{G}_{I \to G}(z_I)) - z_I\|_1]$$
$$+ \mathbb{E}_{z_G \sim p_G}[\|\mathcal{G}_{I \to G}(\mathcal{G}_{G \to I}(z_G)) - z_G\|_1]$$

3. **Alignment Loss**: For paired samples $(z_I^p, z_G^p)$ from the set of paired examples $\mathcal{P}$, embeddings are explicitly aligned in the shared latent space:

$$\mathcal{L}_{align} = \mathbb{E}_{(z_I^p, z_G^p) \sim \mathcal{P}}[\|E_{sh}(z_I^p) - E_{sh}(z_G^p)\|_2^2]$$

The total loss combines these terms with weighting factors:

$$\mathcal{L}_{total} = \lambda_1 \mathcal{L}_{cycle} + \lambda_2 \mathcal{L}_{WGAN} + \lambda_3 \mathcal{L}_{align}$$

Formally, Stage 1 optimization follows a min–max formulation:

$$\min_{\mathcal{G}_{I \to G}, \mathcal{G}_{G \to I}} \left(\lambda_1 \mathcal{L}_{cycle} + \lambda_3 \mathcal{L}_{align}\right) + \min_{\mathcal{G}_{I \to G}, \mathcal{G}_{G \to I}} \max_{\mathcal{D}_I, \mathcal{D}_G} \lambda_2 \mathcal{L}_{WGAN}.$$

During Stage 1, the generators $\mathcal{G}_{I \to G}$ and $\mathcal{G}_{G \to I}$ are updated using $\mathcal{L}_{total}$, while the discriminators $\mathcal{D}_I$ and $\mathcal{D}_G$ are updated using only $\mathcal{L}_{WGAN}$.

**Stage 2: Linear Probing** In the second stage, we train a linear probing head on the paired data. We keep the generators $\mathcal{G}_{I \to G}, \mathcal{G}_{G \to I}$ frozen. These generators are used solely to produce embeddings for each gene expression and image sample in the shared latent space. The resulting embeddings from both modalities are then concatenated and passed to the linear probing head for classification or regression:

$$\hat{y} = h_\theta([E_{sh}(E_I(z^I)); E_{sh}(E_G(z^G))])$$

Only the parameters $\theta$ of the linear probing head are updated during this stage, using a task-specific loss – mean squared error (MSE) for regression or cross-entropy for classification. Definitions of the macro F1 score, micro F1 score, and $R^2$ metric used for evaluation are provided in Appendix F. CAMEO provides a stable, deterministic embedding that accommodates limited paired data and non-coinciding manifold structures (see Appendix G).

## 4 EXPERIMENTS

In this section, we evaluate each multimodal framework's ability to learn from partially paired, small-scale data. Our experiments address two key questions: (1) whether these representations outperform unimodal approaches that leverage only image or gene expression data, and (2) how CAMEO compares to multimodal baselines that require larger amounts of paired data. We assess performance across a wide range of encoder combinations, varying both gene expression and vision backbones in a plug-and-play manner. In addition, we examine how performance changes under different pairing ratios, ranging from fully paired to extremely limited paired data, allowing us to isolate the impact of cross-modal alignment under realistic constraints. We describe the datasets in Section 4.1, the tasks in Section 4.2, the baselines in Section 4.3, and report quantitative results spanning three tissues (breast, lung, and thymus) in Section 4.4. Throughout, we emphasize the role of encoder choice, data availability, and task characteristics in shaping the success of multimodal learning strategies.

### 4.1 DATA

For the evaluation of multimodal learning frameworks and unimodal baselines, we use two 10x Xenium and one 10x Visium spatial transcriptomics cohorts. Each dataset includes expert-generated ground-truth annotations, but predicting these labels is challenging due to intra-class heterogeneity, class imbalance, and the need to account for spatial and contextual dependencies. This makes them well-suited for methods that integrate complementary information from two independent modalities. The Lung Pulmonary Fibrosis (LungPF) dataset comprises 23 10x Xenium samples (Vannan et al., 2023), collected to investigate the pathobiology of PF. A second 10x Xenium cohort includes 7 publicly available breast cancer samples from 4 patients, hosted on the 10x Genomics website and annotated by our collaborating pathologist. The third dataset is a 10x Visium cohort of 19 thymus samples (Yayon et al., 2024) from 11 donors (6 pediatric, 5 fetal), used to study T cell development during fetal and early postnatal stages. To construct HuggingFace (HF) datasets for multimodal learning, we aligned gene expression and imaging modalities by linking each image patch to its corresponding gene expression profile. All datasets will be shared with the research community. We split each into training, validation, and test sets, stratified by patient (see Appendix I).

### 4.2 DOWNSTREAM TASKS

#### 4.2.1 NICHE CLASSIFICATION

Niches are biologically relevant tissue regions that regulate cell behavior, typically involving multiple cell types and extracellular components. Niche annotation involves identifying these regions based on expert-labeled ground truth, enabling detailed tissue characterization. This is crucial for understanding region-specific processes, such as immune infiltration or stromal activity, which are linked to prognosis and treatment outcomes. In this task, image patches are treated as niches and classified into $n_{classes}$ different classes based on tissue morphology, as annotated by a pathologist. For the LungPF cohort, 24 histological niches were identified, each representing a distinct pulmonary fibrotic tissue region. Similarly, for the breast cohort, 14 distinct histological features were annotated. For the thymus cohort, 11 regions were annotated based on morphological landmarks (Yayon et al., 2024). Further details and distributions of the annotations are shown in Appendix I.

#### 4.2.2 CELL TYPE COMPOSITION PREDICTION

Many diseases, including cancer and autoimmune disorders, involve changes in cell type composition. Estimating cell type composition from gene expression data enables the identification of disease-associated diagnostic bio-markers and provides insight into the disease microenvironment, which is crucial for developing targeted therapies. In the cell type composition prediction task, the goal is to estimate the fraction of each cell type present in a patch. We evaluate our model on predicting the composition of 9 broad cell types defined by the group that created the LungPF cohort. Further details are available in Appendix I.

Table 1: Multi-class niche classification performance comparison of models on the Lung ($n_{classes} = 24$), Breast ($n_{classes} = 14$), and Thymus ($n_{classes} = 11$) datasets using **100% paired data**. An asterisk (*) denotes that the model was not evaluated on the test set. The GAT model is only compatible with single-cell spatial data. As a result, we exclude the GAT encoder from experiments on the Thymus dataset, which contains spot-based gene expression measurements. Blue, orange, and green text indicate image encoders, gene expression encoders, and multimodal methods, respectively.

| Data Type | Model Name | Lung | | Breast | | Thymus | |
| --- | --- | --- | --- | --- | --- | --- | --- |
| | | Macro F1 | Micro F1 | Macro F1 | Micro F1 | Macro F1 | Micro F1 |
| Image | CONCH | 0.099 | 0.286 | 0.169 | 0.197 | 0.152 | 0.211 |
| | CTransPath | 0.105 | 0.304 | 0.128 | 0.188 | 0.127 | 0.218 |
| | ResNet | 0.096 | 0.246 | 0.104 | 0.156 | 0.136 | 0.205 |
| | UNI | 0.134 | 0.401 | 0.133 | 0.133 | 0.127 | 0.194 |
| | Virchow | 0.116 | 0.391 | 0.145 | 0.146 | 0.158 | 0.222 |
| Gene Expression | CellPLM | 0.183 | 0.199 | 0.130 | 0.369 | 0.134 | 0.227 |
| | GAT | 0.171 | 0.263 | 0.181 | 0.383 | * | * |
| | Nicheformer | 0.158 | 0.211 | 0.134 | 0.242 | 0.102 | 0.183 |
| | scGPT | 0.177 | 0.269 | 0.153 | 0.386 | 0.134 | 0.208 |
| | scVI | 0.125 | 0.234 | 0.144 | 0.312 | 0.164 | 0.253 |
| Multimodal | CAMEO-CTransPath-GAT | 0.166 | 0.177 | 0.096 | 0.335 | * | * |
| | CAMEO-CTransPath-Nicheformer | 0.075 | 0.064 | 0.080 | 0.233 | 0.193 | **0.347** |
| | CAMEO-CTransPath-scVI | 0.158 | 0.223 | 0.079 | 0.294 | 0.155 | 0.214 |
| | CAMEO-CONCH-GAT | 0.128 | 0.287 | 0.124 | 0.374 | * | * |
| | CAMEO-CONCH-Nicheformer | 0.092 | 0.253 | 0.186 | 0.277 | 0.207 | 0.290 |
| | CAMEO-CONCH-scVI | 0.164 | 0.278 | 0.154 | **0.445** | 0.212 | 0.293 |
| | CAMEO-UNI-GAT | 0.137 | 0.274 | 0.098 | 0.227 | * | * |
| | CAMEO-UNI-Nicheformer | 0.193 | 0.308 | 0.141 | 0.297 | 0.172 | 0.302 |
| | CAMEO-UNI-scVI | 0.137 | 0.249 | 0.181 | 0.411 | 0.185 | 0.267 |
| | CLIP-CTransPath-GAT | 0.199 | **0.412** | 0.204 | 0.407 | * | * |
| | CLIP-CTransPath-Nicheformer | 0.188 | 0.330 | 0.225 | 0.382 | 0.176 | 0.255 |
| | CLIP-CTransPath-scVI | 0.182 | 0.352 | 0.209 | 0.419 | 0.202 | 0.269 |
| | CLIP-CONCH-GAT | 0.180 | 0.358 | 0.195 | 0.341 | * | * |
| | CLIP-CONCH-Nicheformer | 0.158 | 0.309 | 0.196 | 0.341 | 0.173 | 0.247 |
| | CLIP-CONCH-scVI | 0.157 | 0.315 | 0.184 | 0.342 | 0.161 | 0.248 |
| | CLIP-UNI-GAT | 0.195 | 0.382 | 0.194 | 0.422 | * | * |
| | CLIP-UNI-Nicheformer | 0.177 | 0.334 | 0.133 | 0.332 | 0.170 | 0.251 |
| | CLIP-UNI-scVI | 0.194 | 0.335 | 0.170 | 0.305 | 0.188 | 0.259 |
| | Concat-CTransPath-GAT | **0.214** | 0.368 | 0.293 | 0.337 | * | * |
| | Concat-CTransPath-Nicheformer | 0.203 | 0.399 | 0.218 | 0.266 | 0.187 | 0.283 |
| | Concat-CTransPath-scVI | 0.165 | 0.300 | 0.240 | 0.241 | **0.227** | 0.301 |
| | Concat-CONCH-GAT | 0.162 | 0.350 | **0.305** | 0.359 | * | * |
| | Concat-CONCH-Nicheformer | 0.181 | 0.298 | 0.140 | 0.187 | 0.176 | 0.276 |
| | Concat-CONCH-scVI | 0.108 | 0.277 | 0.250 | 0.300 | 0.179 | 0.253 |
| | Concat-UNI-GAT | 0.145 | 0.392 | 0.152 | 0.233 | * | * |
| | Concat-UNI-Nicheformer | 0.130 | 0.398 | 0.140 | 0.141 | 0.148 | 0.198 |
| | Concat-UNI-scVI | 0.126 | 0.322 | 0.136 | 0.199 | 0.152 | 0.190 |

## 4.3 BASELINES

We evaluate three categories of baselines: image-only, gene expression-only, and multimodal. Image baselines use five pretrained encoders: UNI (Chen et al., 2024), CTransPath (Wang et al., 2022), ResNet-50 (He et al., 2015), Virchow (Vorontsov et al., 2024), and CONCH (Lu et al., 2024a). These models are used in a frozen state, and a linear classifier is trained on top (linear probing) to assess the quality of their representations. Gene expression baselines take high-dimensional gene expression vectors as input. We apply linear probing to CellPLM (Wen et al., 2023), scGPT (Cui et al., 2024a), Nicheformer (Schaar et al., 2024), and the GAT encoder described in Section 3.3, keeping all encoders frozen. For scVI (Lopez et al., 2018), we train the encoder from scratch using only the training split of our paired HuggingFace datasets, and apply a linear classifier on top of the learned embeddings. For multimodal baselines, we consider two frameworks: CLIP (Radford et al., 2021) and "Concat". In the "Concat" framework, the raw output embeddings from the unimodal gene expression and image encoders are concatenated directly, without any explicit alignment or fusion mechanism. We test each multimodal framework using different combinations of gene expression and image encoders. All multimodal frameworks produce joint embeddings that are used to train a task-specific linear head. Further details for all baselines are provided in Appendix E.

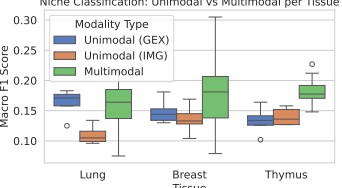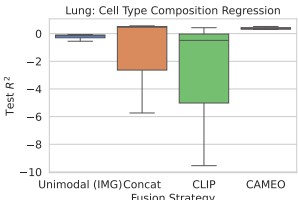

Figure 2: Performance comparison by modality and fusion strategy. Left: Encoder combinations used in each fusion method across tissues. Center: Macro F1 scores for unimodal versus multimodal models using 100% paired data. Right: Regression test $R^2$ for cell type composition prediction, showing multimodal models outperform unimodal approaches, with CAMEO achieving the highest performance.

## 4.4 RESULTS

All models were evaluated using the same train, validation, and test splits (stratified by patient). Due to significant class imbalances (shown in Appendix I), we use the macro F1 score as our primary metric for niche classification, and the $R^2$ score for cell type composition regression.

Our experiments across three tissue types (Lung, Breast, and Thymus) yielded several consistent findings. First, multimodal approaches outperform their unimodal counterparts, demonstrating the value of integrating complementary information from images and gene expression data, as shown in Fig. 2 (center). This advantage is further validated in our cell type composition prediction task, where CAMEO achieves the highest $R^2$ scores (Fig. 2, right). However, the performance gains vary across tissue types and encoder combinations, as illustrated in Fig. 2 (left). In the Lung dataset, Concat–CTransPath–GAT achieved the highest macro F1 score (0.214). For Breast tissue, Concat–CONCH–GAT performed best (0.305), while in Thymus, Concat–CTransPath–scVI yielded the top result (0.227). Interestingly, CAMEO outperformed all Concat variants in terms of micro F1 score for both Breast and Thymus, achieving 0.445 with CAMEO–CONCH–scVI in Breast and 0.347 with CAMEO–CTransPath–Nicheformer in Thymus (Table 1). While Concat models are surprisingly strong when trained on 100% paired data (in terms of macro F1), their performance declines as the proportion of paired data decreases, as we show next.

The relationship between pairing percentage and performance reveals that while performance typically

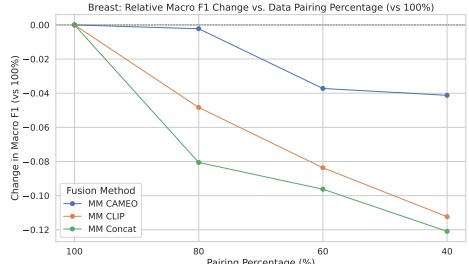

Figure 3: Relative change in Breast macro F1 scores with decreasing pairing percentage. CAMEO shows greater robustness compared to CLIP and Concat. A similar patterns for Lung can be seen in App. Fig. 5.

Table 2: Niche classification performance under reduced data pairing scenarios using default UNI and scVI encoders.

| Tissue | Method | 100% | 80% | 60% | 40% ($\triangle$) |
|--------|--------|------|-----|-----|-----------|
| Lung | CAMEO | 0.137 | **0.153** | **0.158** | **0.143** ($\triangle$ **4.4%**) |
| | CLIP | **0.194** | 0.146 | 0.128 | 0.133 ($\triangle$ -31.5%) |
| | Concat | 0.126 | 0.125 | 0.117 | 0.123 ($\triangle$ -2.2%) |
| Breast | CAMEO | **0.181** | **0.157** | 0.086 | **0.103** ($\triangle$ -43.3%) |
| | CLIP | 0.170 | 0.103 | 0.108 | 0.078 ($\triangle$ -54.4%) |
| | Concat | 0.136 | 0.119 | **0.122** | 0.102 ($\triangle$ **-25.1%**) |

decreases with less paired data, the rate of decline varies significantly between approaches. Notably, CAMEO demonstrates greater robustness to reduced pairing percentages than CLIP, particularly in the Breast dataset (Fig. 3). Similar trends are observed in the Lung dataset (App. Fig. 5). This robustness pattern is consistent across both classification and regression tasks, with CAMEO maintaining stronger performance as paired data decreases (App. Fig. 5). These trends reflect mean performance across encoder combinations. Table 2 provides detailed results using the UNI and scVI encoders: for the lung dataset, CAMEO shows a 4.4% improvement at 40% pairing compared to 31.5% and 2.2% drops for CLIP and Concat, respectively; for Breast tissue, CAMEO maintains the highest performance (0.103) despite reduced pairing. We attribute CAMEO's robustness to its adversarial alignment procedure, which effectively leverages unpaired data to preserve representation quality.

### 4.5 ABLATION STUDIES

#### 4.5.1 ENCODER SELECTION IMPACT VARIES ACROSS FRAMEWORKS

Our results reveal that encoder selection critically impacts performance, but optimal combinations differ across multimodal frameworks. As an example, in the Lung dataset, the best-performing CAMEO variant is CAMEO–UNI–Nicheformer (0.193), whereas Concat achieves its highest score with Concat–CTransPath–GAT (0.214). This inconsistency extends to other tissues as well, where the best-performing encoder pair for one framework rarely aligns with the optimal choice for another (Table 1). This finding underscores the importance of comprehensive encoder evaluations when implementing any multimodal approach.

#### 4.5.2 UNIMODAL PERFORMANCE DOES NOT PREDICT MULTIMODAL SUCCESS

A surprising finding is that the performance of a unimodal encoder poorly predicts its effectiveness in multimodal settings, as illustrated in Fig. 4 for the Breast dataset. For example, while GAT achieves the highest macro F1 score (0.181) among gene expression encoders for this dataset, the best-performing multimodal combination in CLIP and CAMEO uses Nicheformer instead. This pattern persists across tissues (App. Fig. 6), suggesting that complementarity between encoders may be more important than individual encoder strength. This non-predictive relationship challenges the common practice of selecting encoders based solely on their standalone performance and highlights the need for systematic evaluation of encoder combinations in multimodal frameworks.

### 4.6 RECOMMENDATIONS

Based on our findings, we recommend the following best practices for developing multimodal frameworks: (i) systematically evaluate multiple encoder combinations, as the optimal pairing of gene expression and image encoders can vary across both tissue types and multimodal integration methods; (ii) include concatenation as a competitive baseline, since it often performs well; and (iii) consider adversarial methods like CAMEO when working with limited paired data or in settings with label imbalance, as they can effectively exploit unpaired examples.

## 5 DISCUSSION

Our study introduces adversarial alignment as an effective alternative to existing multimodal fusion approaches in biomedical settings where paired samples are scarce. CAMEO's superior robustness to reduced paired data demonstrates the value of adversarial mechanisms for leveraging unpaired examples, which is particularly valuable in computational pathology, where unimodal data is abundant but generating paired datasets remains challenging. Unexpectedly, we found that the performance of unimodal encoders is a poor predictor of multimodal success, challenging conventional wisdom in model selection and highlighting the importance of encoder comple-

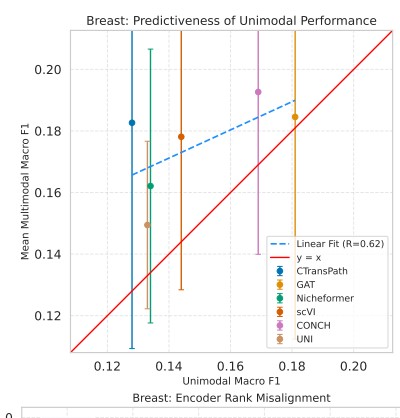

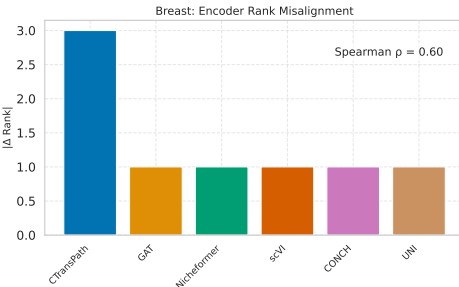

Figure 4: Predictiveness of unimodal performance for multimodal success in the Breast dataset. Moderate correlation (Spearman $\rho = 0.60$) and high variance across multimodal models, and substantial rank misalignments (bottom panel). Results for Lung and Thymus are shown in App. Fig. 6.

mentarity. This suggests that the way encoders interact within a fusion framework may be more important than their individual strengths, requiring evaluation of multiple combinations when implementing multimodal approaches. While our work focused on spatial omics and histopathology, the adversarial alignment principles explored here could extend to other domains with limited paired data. By releasing expert-annotated multimodal datasets and providing practical insights for modality fusion strategies, we contribute both methodological advancement and resources to facilitate further exploration of multimodal learning in data-constrained environments.

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

## A    DATA AND CODE AVAILABILITY

All data and code necessary to reproduce the reported results will be released upon publication.

## B    LLM USAGE

We used a large language model (LLM) to assist with text editing and figure formatting. The LLM was used only for improving clarity and presentation; all scientific content, analyses, and interpretations were generated by the authors.

## C    LIMITATIONS

CAMEO builds on the hypothesis that two modalities can provide distinct but complementary information. This assumption introduces certain limitations:

- **Representation Quality.** CAMEO aligns representations learned by self-supervised uni-modal encoders. Its ability to meaningfully align these representations for downstream tasks is limited by the quality of the unimodal representations. This limitation is an opportunity at the same time, as the modular design allows the user to replace the encoders with improved models.
- **Modality Translation.** The assumption of complementarity limits the alignment of modalities that do not share common concepts. If one modality, for example, is independent of the other, the translation becomes random.
- **Downstream Task Suitability.** Our approach was tested on computational pathology tasks chosen for their inherently multimodal nature, where both histology and gene expression data are informative. This approach might not make sense for every task. Specifically, it may not generalize to tasks for which one modality yields little or no added benefit.
- **Loss of Modality-Specific Information.** The alignment process may reduce modality-specific information, as it prioritizes shared representations. A potential future direction is to learn both shared and modality-specific latent factors, preserving information unique to each modality while still benefiting from their alignment.

## D    ADDITIONAL RESULTS

This section provides additional results and visualizations that complement the main findings presented in Section 4.4.

### D.1    IMPACT OF REDUCED PAIRING PERCENTAGES

Fig. 5 shows the relative performance changes for the Lung dataset as the percentage of paired data decreases, complementing the Breast dataset results presented in Fig. 3.

### D.2    PREDICTIVENESS OF UNIMODAL PERFORMANCE

Fig. 6 extends the analysis presented in Fig. 4 to the Lung and Thymus datasets. The weak correlations (Spearman $\rho$ values of 0.37 and 0.31, respectively) confirm that the pattern observed in the Breast dataset ($\rho = 0.60$) is consistent across tissues, though with varying strengths.

## E    MODEL DETAILS

### E.1    IMAGE BASELINES

The image baselines leverage pretrained encoders to extract embeddings from histopathological images. These encoders are used in a frozen state, with a task-specific head trained on top.

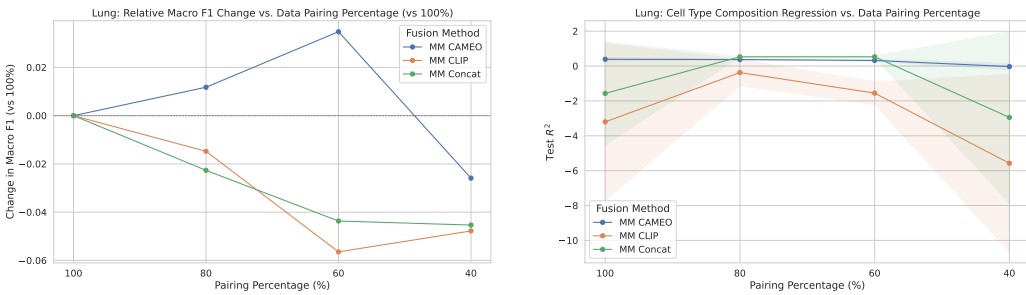

Figure 5: Lung dataset: Relative change in macro F1 scores compared to 100% pairing (left). Lung shows similar patterns to Breast (Fig. 3). Mean and standard deviation of absolute $R^2$ scores for cell type composition prediction (right).

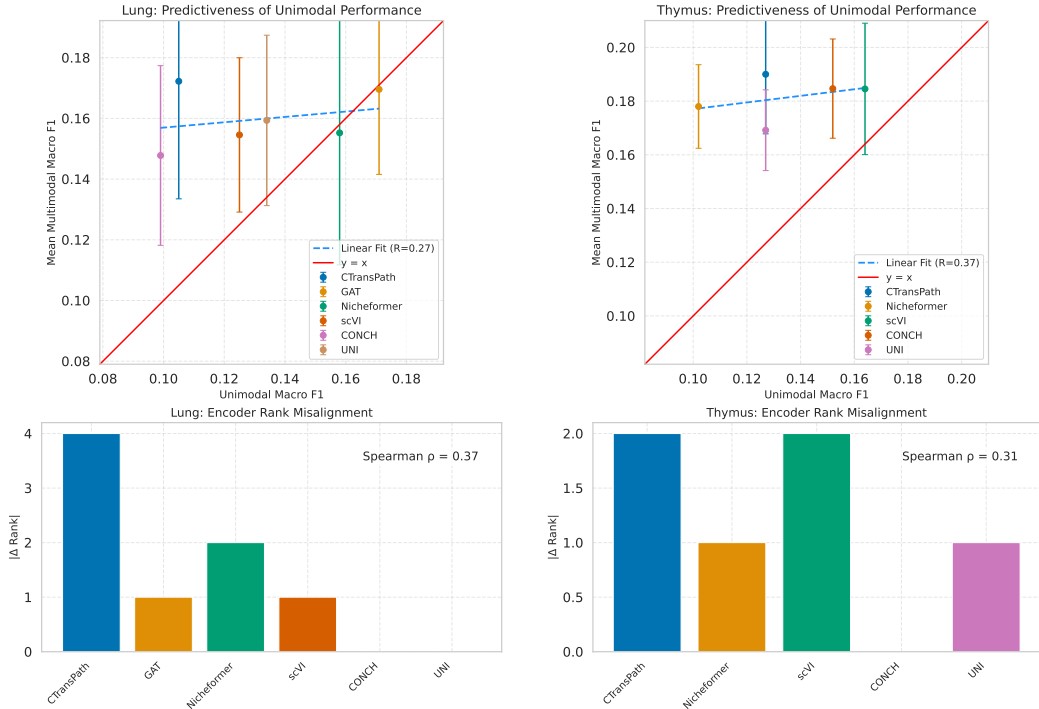

Figure 6: Predictiveness of unimodal performance for multimodal success in Lung (left) and Thymus (right) datasets. Both tissues show weak correlations ($\rho = 0.37$ and $\rho = 0.31$, respectively) and significant rank misalignments, further supporting the finding that unimodal performance poorly predicts multimodal success.

- **UNI** (Chen et al., 2024): UNI is built on a ViT-Large (ViT-L) architecture and pretrained on over 100 million image patches derived from 100,000 H&E-stained whole-slide images (WSIs) spanning 20 major tissue types.
- **CTransPath** (Wang et al., 2022): Integrates a convolutional neural network with a multi-scale Swin Transformer architecture. It is pretrained on 15 million unlabeled image patches from The Cancer Genome Atlas (TCGA) and PAIP datasets.
- **ResNet-50** (He et al., 2015): A convolutional neural network pre-trained on ImageNet-1k.
- **Virchow** (Vorontsov et al., 2024): A self-supervised vision transformer pretrained on 3.1 million whole-slide histopathology images.
- **CONCH** (Lu et al., 2024a): A vision-language model trained via contrastive learning on 1.17 million histopathology-specific image–caption pairs.

## E.2 GENE EXPRESSION BASELINES

Gene expression baselines process high-dimensional omics data, where each sample is represented as a vector of gene expression values.

- **GAT**: Graph Attention Networks (GATs) (Velickovic et al., 2017) learn node representations by dynamically adjusting the influence of neighboring nodes via attention weights. The attention mechanism computes a weighted aggregation of neighbors' features, where the attention coefficients are learned based on the similarity of node features. This allows the model to focus on more relevant neighbors, making GATs particularly suitable for graph-structured data such as spatial transcriptomics measurements. In this context, $\alpha_{uv}$ represents the attention weight between nodes $u$ and $v$, which is learned as a function of the features of both nodes, modulated by the graph structure $G$. While GATs are typically trained in an unsupervised manner, self-supervised learning (SSL) has been shown to enhance their performance, particularly when applied to large datasets (Richter et al., 2023). The model is pre-trained using masked modeling, where parts of the gene expression profile $x_v$ are masked, and the objective is to predict the masked entries from the unmasked ones. Let $f_\theta(x_v)$ denote the unimodal encoder, which maps the gene expression vector $x_v$ to an embedding $z_v$ used later in the fusion module. The training objective is to minimize the masked modeling loss, which for a set of masked nodes $\mathcal{M}$ is:

$$\mathcal{L}_{\text{SSL}} = \sum_{v \in \mathcal{M}} ||(1_p - M_v) \odot f_\theta(M_v \odot x_v) - \hat{x}_v||_2^2, \tag{1}$$

  where $M_v \in \{0, 1\}^p$ is a mask for the genes of cell $v$ and $(1_p - M_v)$ ensures that only the masked entries contribute to the loss. This SSL pretraining enables the model to learn meaningful representations from large unimodal datasets without requiring labeled data. By capturing complex spatial dependencies in gene expression patterns, the learned embeddings provide a strong foundation for downstream tasks.

- **scVI** (Lopez et al., 2018): A probabilistic framework for analyzing gene expression data. It employs a variational autoencoder (VAE) to learn a low-dimensional latent representation of gene expression profiles.

- **scGPT** (Cui et al., 2024a): Adapts the GPT architecture for gene expression data by treating genes as words (tokens) and cells as sentences. It is pretrained on 33 million cross-tissue human cells.

- **CellPLM** (Wen et al., 2023): A spatial transcriptomics foundation model that explicitly encodes cell spatial relationships. CellPLM is trained on 9 million scRNA-seq cells and 2 million spatial transcriptomics cells.

- **Nicheformer** (Schaar et al., 2024): A transformer-based foundation model tailored for spatial single-cell omics. It is pretrained on 57 million dissociated and 53 million spatially resolved cells from 73 human and mouse tissues.

### E.2.1 MULTIMODAL BASELINES

- **CLIP** (Radford et al., 2021): A multimodal model trained via contrastive learning. Each modality is independently encoded and projected into modality-specific latent spaces. A contrastive loss aligns paired samples, enabling joint representation learning.

- **Concatenated**: Concatenates the embeddings produced by the gene expression and image encoders, without any explicit alignment or fusion mechanism.

## E.3 CAMEO ARCHITECTURE

- **Input Dimensions:**
  - Image embeddings: $d_{\text{img}} = 1024$ (UNI) or $768$ (CTransPath) or $512$ (CONCH)
  - Gene expression embeddings: $d_{\text{gex}} = 128$ (GAT) or $128$ (scVI) or $512$ (Nicheformer)
  - Common latent space: $d = \min(d_{img}, d_{gex})$
- **Generator Architecture:**

- – Encoder layers: $d \rightarrow d \rightarrow d/2$
- – Decoder layers: $d/2 \rightarrow d \rightarrow d$
- – Shared encoder layer: $d/2 \rightarrow d/4$
- – Shared decoder layer: $d/4 \rightarrow d/2$
- – All layers include BatchNorm1d and LeakyReLU ($\alpha = 0.2$)

- **Discriminator Architecture:**
  - – Input layer: $d \rightarrow d/4$
  - – Hidden layer: BatchNorm1d + LeakyReLU
  - – Output layer: $d/4 \rightarrow 1$

## E.4 CAMEO TRAINING CONFIGURATION

- Shared layers use LeakyReLU activation ($\alpha = 0.2$)
- Batch normalization in discriminators
- Weight clipping range: $[-0.01, 0.01]$ for Wasserstein GAN
- Optimizers: RMSprop (Wasserstein) or AdamW (standard GAN)

## E.5 TRAINING PROTOCOLS

**Gene Expression Encoder Pre-training** The GAT model is pre-trained using a masked gene expression prediction task:

- Masking ratio: 0.4 (40% of expression values masked)
- Learning rate: 0.001
- Batch size: 1 (full patch per batch)
- Early stopping patience: 200 epochs
- Maximum epochs: 5000
- Gradient clipping: 1.0
- Loss: MSE or Negative Binomial (when predict_nb=True, Negative Binomial performs better)

**scVI Training** The scVI models for the Lung and Breast datasets were pre-trained using default parameters from the scVI-tools pipeline. For Thymus, the model was pre-trained following the authors' recommendations (Yayon et al., 2024), using sample ID as a batch covariate.

**CLIP and CAMEO Training** For CLIP and CAMEO models, we evaluated a pre-training duration of 10 epochs. For CAMEO specifically, we optimized $\lambda_{align}$ to balance the contribution of the alignment loss term.

**Linear Probing Stage** We adopted a linear probing setup to isolate the quality of the pretrained representations without updating encoder weights. Linear probes were trained using the AdamW optimizer with a learning rate of 1e-3 for 10 epochs. This learning rate was found to perform robustly across different model initializations and tasks.

**CAMEO-specific Parameters**

- $\lambda_{align}$: Controls the weight of alignment loss
- Generator Learning Rate: 1e-4
- Discriminator Learning Rate: 5e-5
- Weight Clipping Value: 0.01

## F EVALUATION METRICS

Let $y \in \{1, \ldots, C\}$ be the true class label and $\hat{y}$ the predicted class label for $N$ samples.

**Macro F1 score.** The F1 score for class $c$ is defined as

$$\text{F1}_c = \frac{2\,\text{Precision}_c \cdot \text{Recall}_c}{\text{Precision}_c + \text{Recall}_c},$$

where

$$\text{Precision}_c = \frac{\text{TP}_c}{\text{TP}_c + \text{FP}_c}, \quad \text{Recall}_c = \frac{\text{TP}_c}{\text{TP}_c + \text{FN}_c}.$$

The macro F1 score averages $\text{F1}_c$ over all $C$ classes with equal weight:

$$\text{MacroF1} = \frac{1}{C}\sum_{c=1}^{C}\text{F1}_c.$$

Macro F1 treats all classes equally and is sensitive to performance on underrepresented classes.

**Micro F1 score.** The micro F1 score computes precision and recall by first aggregating TP, FP, and FN across all classes, and then applying the F1 formula to these aggregated counts. Micro F1 reflects overall model performance weighted by class frequency.

$R^2$ **score.** For regression tasks, the coefficient of determination is defined as

$$R^2 = 1 - \frac{\sum_{i=1}^{N}(y_i - \hat{y}_i)^2}{\sum_{i=1}^{N}(y_i - \bar{y})^2},$$

where $\bar{y}$ is the sample mean of $\{y_i\}_{i=1}^{N}$. $R^2$ measures the proportion of variance explained by the model.

## G   COMPARISON TO PROBABILISTIC ALIGNMENT METHODS

We compare two prominent probabilistic alignment methods in the context of histopathology-gene expression alignment:

**DDMEC** (Bounoua et al., 2025) learns stochastic conditionals between modalities using diffusion models with KL-regularized "soft marginals". The approach offers principled uncertainty modeling but requires significant computational resources for training and sampling from the diffusion models.

**scTopoGAN** (Singh et al., 2023) combines topology-preserving autoencoders with GAN-based latent alignment. The method assumes shared topological structure between modalities - an assumption well-suited for molecular measurements (e.g., ATAC-seq, CITE-seq) but less applicable when modalities capture fundamentally different biological features at distinct scales.

For histopathology-gene expression alignment, key practical considerations include:

1. Computational efficiency for large spatial patches
2. Non-coinciding manifold structure between modalities
3. Availability of paired anchors, even if limited
4. Requirements for stable embeddings in downstream tasks

These constraints favor deterministic approaches like CAMEO for this specific use case, though probabilistic methods remain valuable in settings where their assumptions hold and computational constraints permit.

## H   COMPUTATIONAL RESOURCES

Models were trained using Python 3.10 on a GPU server with these specifications:

- GPU: 30 nodes, each with 96 CPUs, 768GB memory, 4 NVIDIA H100 GPUs each with 94GB HBM2 memory.

- GPU: 5 nodes, each with 96 CPUs, 1TB memory, 4 NVIDIA A100 GPUs each with 80GB HBM2 memory.
- GPU: 4 nodes, each with 252 CPUs, 2TB memory, 8 NVIDIA A100 GPUs each with 80GB HBM2 memory.

Training each model typically took less than 6 hours, and in most cases, less than 2 hours.

# I DATASETS

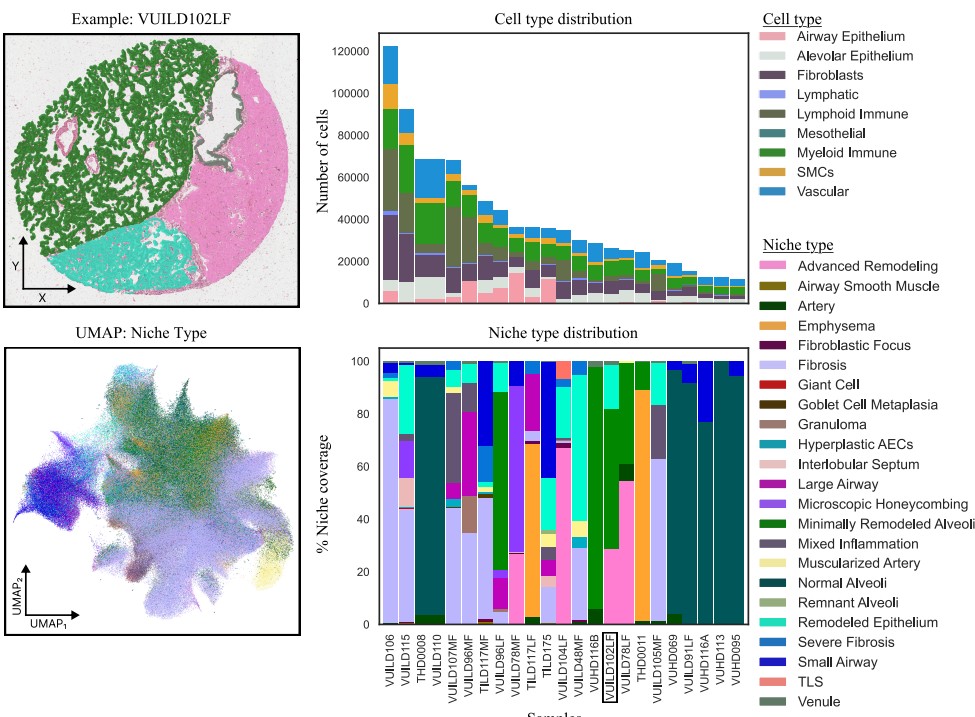

Figure 7: Metadata overview for the LungPF Xenium cohort (Vannan et al., 2023), highlighting cell-type distributions, niche-label distributions, and the imbalance of labels across samples. Top left: example histology image with expert-provided niche annotations overlaid; the corresponding niche-label distribution for this sample is shown in the bar chart (sample name boxed for reference). Top right: total cell-type distribution across all 23 samples, showing slight variability in cell-type composition. Bottom: UMAP of patch-level gene expression profiles and overall niche-label distribution, revealing substantial imbalance across the 23 unique annotated niche types. Niches labeled 'NOANNOT' are excluded from all visualizations.

Paired spatial transcriptomics datasets remain scarce. Often, spatial datasets are released with only the genomic component and its corresponding spatial context, while the paired histology is omitted due to the prohibitive costs of publishing large images. While this is gradually improving, collecting large-scale paired datasets remains challenging. To support research on partially paired spatial data, we created a HuggingFace dataset that combines histology images and gene expression data at the niche level. A niche is defined as a $224 \times 224$ pixel crop of the histology slide, paired with the corresponding gene expression values from cells located within the crop. Additionally, expert-provided niche annotations are included for each pair, along with the total number of cells per spot, as estimated by Xenium's default segmentation.

To ensure proper spatial alignment between the histological and transcriptomic data, we utilized SpatialData (Marconato et al., 2025). The niches are tessellated across each slide without overlap, and we apply a quality control filter to exclude niches containing less than 50% tissue (i.e., those that predominantly contain background). For simplicity and interpretability, transcripts from partially included cells are treated as whole-cell data within the niche.

The lung dataset consists of 23 Xenium samples from 19 patients, covering both healthy tissue and varying severities of pulmonary fibrosis. Niche regions were annotated by expert pathologists on the H&E-stained images, while single-cell annotations were provided at multiple levels of granularity by the original study's authors. For our analysis, we use the broadest cell-type labels. In total, the dataset includes approximately 1 million cells across $\sim 71K$ patches (Table 3). Figure 8a shows the niche label distribution across the training, validation, and test splits, highlighting label imbalance and distribution shift between the lung samples.

The breast cancer dataset was generated and provided by 10x Genomics, the manufacturer of the Xenium technology. It includes 7 Xenium samples from 4 patients, capturing breast tissue affected by cancer. This dataset comprises $\sim 2.3$ million cells distributed across $\sim 126K$ patches. Niche-level annotations were provided by our collaborating pathologist using the Elston-Ellis scoring system (ELSTON & ELLIS, 1991). Figure 8b shows the niche label distribution across the training, validation, and test splits in the breast cohort.

The thymus dataset consists of 19 Visium samples from 11 donors, spanning fetal (post-conception weeks 11–21) and pediatric (neonate to 3 years old) tissue. This dataset was created to map T cell development in pre- and early postnatal stages. The dataset was annotated using the Cortico-Medullary Axis (CMA), a common coordinate framework developed by the dataset's original authors. As the thymus dataset is a spot-based Visium dataset, it contains transcriptomes from spots rather than individual cells. These spots are distributed across $\sim 40K$ patches. Pediatric tissue samples were used for training, while fetal samples were held out for testing (Table 3). Figure 8c shows the niche label distribution across the training, validation, and test splits in the thymus cohort.

An overview of the dataset composition across all three cohorts is presented in Table 3, which details the number of niches, cell types, cell counts, number of patches, and split assignment per sample. Figure 7 further illustrates the distribution of niche types and their corresponding cell type compositions in the lung cohort, highlighting the heterogeneity within the dataset.

## J  IMPACT STATEMENT

This work advances multimodal learning techniques for scenarios with limited paired data. By releasing our annotated HuggingFace dataset and open-source implementation, we aim to facilitate reproducibility in multimodal research. While our framework is developed for biomedical applications, the underlying techniques could benefit other domains where data pairing is costly or impractical. Given the medical context of our work and the sensitive nature of patient-derived molecular and histological data, applications must carefully consider data privacy, informed consent, and clinical validation requirements. The integration of multiple data modalities further increases the identifiability of patient information, emphasizing the need for responsible data handling.

| Tissue | Patient | Sample | #Niches | #Cell Types | #Cells | #Patches | Split |
|---|---|---|---|---|---|---|---|
| Lung | THD0008 | THD0008 | 5 | 9 | 117,612 | 4,316 | Train |
| Lung | THD0011 | THD0011 | 5 | 9 | 53,544 | 2,099 | Val |
| Lung | TILD117 | TILD117MF | 11 | 9 | 86,728 | 3,831 | Val |
| Lung | TILD117 | TILD117LF | 8 | 9 | 65,423 | 2,800 | Val |
| Lung | TILD175 | TILD175 | 11 | 10 | 59,824 | 2,489 | Train |
| Lung | VUHD069 | VUHD069 | 4 | 9 | 36,862 | 1,673 | Test |
| Lung | VUHD095 | VUHD095 | 4 | 9 | 19,631 | 808 | Train |
| Lung | VUHD113 | VUHD113 | 2 | 9 | 20,850 | 1,281 | Train |
| Lung | VUHD116 | VUHD116B | 4 | 9 | 48,213 | 1,898 | Train |
| Lung | VUHD116 | VUHD116A | 3 | 9 | 19,005 | 926 | Train |
| Lung | VUILD102 | VUILD102LF | 5 | 10 | 45,545 | 2,616 | Train |
| Lung | VUILD104 | VUILD104LF | 11 | 9 | 63,792 | 2,593 | Train |
| Lung | VUILD105 | VUILD105MF | 6 | 9 | 39,419 | 1,779 | Train |
| Lung | VUILD106 | VUILD106 | 11 | 10 | 235,514 | 9,923 | Train |
| Lung | VUILD107 | VUILD107MF | 11 | 9 | 115,596 | 4,392 | Train |
| Lung | VUILD110 | VUILD110 | 5 | 9 | 117,612 | 4,316 | Train |
| Lung | VUILD115 | VUILD115 | 13 | 10 | 176,255 | 7,305 | Train |
| Lung | VUILD48 | VUILD48MF | 8 | 9 | 52,956 | 2,352 | Train |
| Lung | VUILD78 | VUILD78LF | 5 | 9 | 55,614 | 2,227 | Test |
| Lung | VUILD78 | VUILD78MF | 7 | 10 | 73,135 | 2,722 | Test |
| Lung | VUILD91 | VUILD91LF | 5 | 10 | 23,427 | 1,450 | Train |
| Lung | VUILD96 | VUILD96LF | 8 | 9 | 76,223 | 3,479 | Train |
| Lung | VUILD96 | VUILD96MF | 7 | 9 | 95,241 | 4,034 | Train |
| Breast | Patient 1 | Rep1 | 9 | 12 | 261,661 | 11,122 | Val |
| Breast | Patient 1 | Rep2 | 8 | 12 | 170,749 | 7,381 | Val |
| Breast | Patient 2 | Sample 2 | 10 | 16 | 221,766 | 9,101 | Test |
| Breast | Patient 3 | TENX94 | 7 | * | 545,980 | 18,042 | Train |
| Breast | Patient 3 | TENX96 | 7 | * | 557,757 | 18,277 | Train |
| Breast | Patient 4 | TENX95 | 7 | * | 917,241 | 31,154 | Train |
| Breast | Patient 4 | TENX97 | 7 | * | 919,345 | 31,693 | Train |
| Thymus | Z1 | TA11486161 | 11 | * | * | 1,445 | Train |
| Thymus | Z2 | TA11556492 | 11 | * | * | 3,045 | Train |
| Thymus | Z2 | TA11486164 | 11 | * | * | 2,827 | Train |
| Thymus | Z4 | TA11486162 | 11 | * | * | 3,210 | Train |
| Thymus | Z4 | TA11556493 | 11 | * | * | 2,963 | Train |
| Thymus | Z4 | TA11556494 | 11 | * | * | 2,481 | Train |
| Thymus | Z6 | TA11556496 | 11 | * | * | 2,295 | Train |
| Thymus | Z6 | TA11556495 | 11 | * | * | 2,764 | Train |
| Thymus | F113 | WSSS_F_IMMsp11604689 | 10 | * | * | 311 | Val |
| Thymus | F114 | WSSS_F_IMMsp11604687 | 11 | * | * | 1,264 | Test |
| Thymus | F114 | WSSS_F_IMMsp11604688 | 11 | * | * | 1,162 | Test |
| Thymus | F130 | WSSS_F_IMMsp11604686 | 11 | * | * | 2,121 | Test |
| Thymus | F130 | WSSS_F_IMMsp11604685 | 11 | * | * | 1,846 | Test |
| Thymus | F133 | WSSS_F_IMMsp11765867 | 11 | * | * | 1,606 | Val |
| Thymus | F135 | WSSS_F_IMMsp11765870 | 11 | * | * | 2,570 | Val |
| Thymus | U09 | WSSS_THYst9518033 | 11 | * | * | 2,125 | Train |
| Thymus | U11 | WSSS_THYst9518030 | 11 | * | * | 2,384 | Train |
| Thymus | U11 | WSSS_THYst9142088 | 11 | * | * | 2,035 | Train |
| Thymus | U11 | WSSS_THYst9142089 | 11 | * | * | 2,190 | Train |

Table 3: Summary of samples, patients, unique niche annotations (including NOANNOT), unique cell types (including NOMAP), cell counts, number of patches, and data split (train/val/test) for the lung, breast and thymus datasets. Individual cells can be counted more than once when they intersect with multiple patches. For the thymus dataset, which is spot-based rather than cell-based, cell counts and cell-type classifications do not apply and are denoted with *. Similarly, cell-type annotations are unavailable for certain breast samples; in those cases, the number of unique cell types is also marked with *.

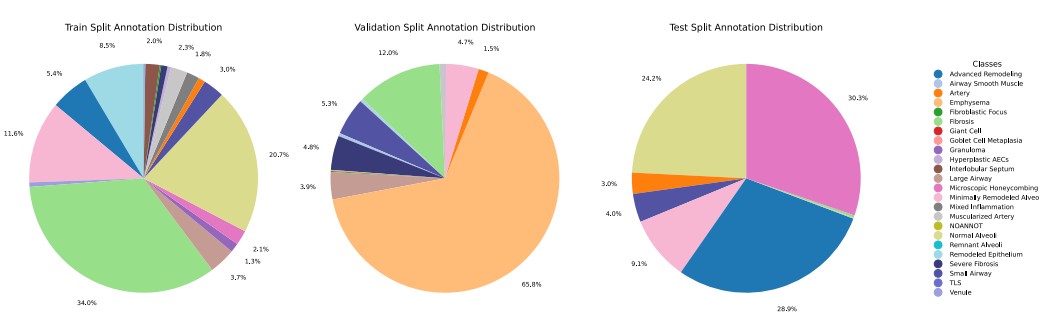

(a) Label distribution in the lung cohort across train, validation, and test splits, defined based on patient stratification. The NOANNOT category is excluded from the visualization. A large distribution shift is observed between the splits.

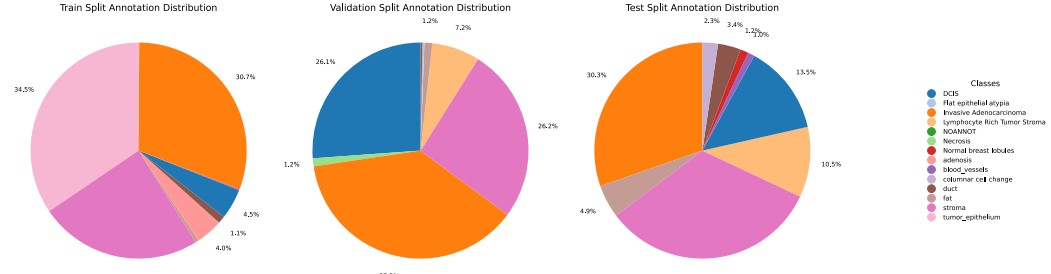

(b) Label distribution in the breast cohort across train, validation, and test splits, defined based on patient stratification. The NOANNOT category is excluded from the visualization. A moderate distribution shift is observed between the splits.

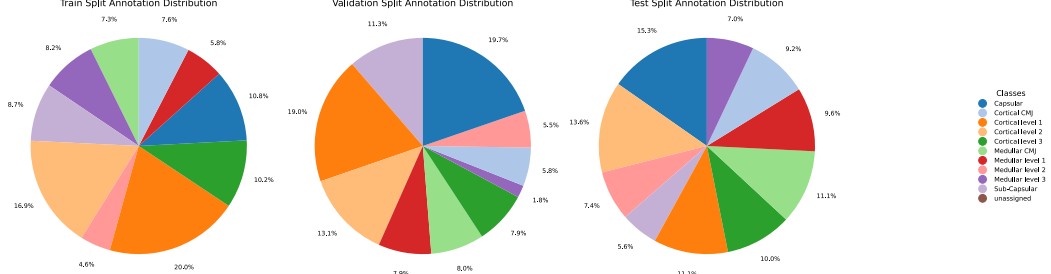

(c) Label distribution in the thymus cohort across train, validation, and test splits, defined based on patient stratification. The NOANNOT category is excluded from the visualization. A small distribution shift is observed between the splits.

Figure 8: Label distributions in the lung, breast, and thymus cohorts across train, validation, and test splits (patient-stratified). The NOANNOT category is excluded.