# OpenReview forum: "Learning Multimodal Representations from Partially Paired, Small-Scale Data"
_ICLR.cc/2026/Conference — ICLR 2026 Conference Withdrawn Submission_

### Official Review · Reviewer_LbuY · 2025-10-26

**Soundness:** 2
**Presentation:** 2
**Contribution:** 2
**Rating:** 2
**Confidence:** 4

**Summary:**

This paper introduces CAMEO, an adversarial learning framework that learns multimodal representations by aligning pre-trained unimodal encoders to address the challenge of scarce paired data in biomedicine. Experiments demonstrate that CAMEO is more robust than baselines like CLIP in low-data regimes and reveal that complementarity between encoders is more critical for successful multimodal fusion than their standalone performance.

**Strengths:**

1. The proposed CAMEO framework could leverage abundant unpaired data through its adversarial alignment mechanism, making it suitable for real-world scenarios where paired data is scarce.

2. The curation and release of a new, expert-annotated, patch-level multimodal dataset is a contribution that will facilitate reproducible research and future benchmarking.

**Weaknesses:**

1.Overly Broad Title and Limited Generalizability: The title, "LEARNING MULTIMODAL REPRESENTATIONS FROM PARTIALLY PAIRED, SMALL-SCALE DATA," suggests a general-purpose method. However, all experiments are exclusively conducted within the biomedical domain.

2.Misalignment Between Stated Problem and Experimental Focus: The paper's primary stated goal is to address the problem of partially paired multimodal representation learning. However, a substantial portion of the experimental results and discussion is devoted to the separate issue of encoder selection. It dilutes the focus on the core contribution and the central challenge of learning from incomplete pairings.

3.Lack of Comparison to Relevant Baselines: The paper fails to compare its method against several important baselines specifically designed for partially paired or missing modality settings. A comprehensive evaluation should include comparisons with methods like Propensity Score Alignment[1], MedFuse[2], and DrFuse[3] to properly contextualize the novelty and performance of CAMEO.

4.Unclear Paper Structure: The organization of the paper could be improved for clarity. For example, the second paragraph of Section 2.3 does not align with its heading ("MODAL ALIGNMENT AND INTEGRATION TECHNIQUES"). The first part of this paragraph describes the authors' own experimental setup, while the second part repeats the contribution summary from the introduction, making the narrative disjointed.

5.Limited Novelty in Technical Contribution: While CAMEO utilizes adversarial learning to handle unpaired data, the three loss functions employed (adversarial, cycle consistency, alignment) are standard components in their respective fields, thus limiting the technical novelty. Furthermore, the paper lacks an ablation study to demonstrate the necessity and individual contribution of each of these three loss components.

6.Findings Lack Novel Insight: The third and fourth claimed contributions—that multimodal approaches outperform unimodal ones and that unimodal performance poorly predicts multimodal success—are well-established findings in the broader multimodal learning literature. The paper re-validates these conclusions in a specific biomedical context but does not offer new insights into why these phenomena occur within their framework or domain.


7.Inference-Time Limitation: The proposed CAMEO framework appears to be limited in its practical application. While it can leverage partially paired data during the training phase, it seems to require fully paired data at the test/inference stage, as both modalities are needed for the final prediction. This could restrict its utility in real-world scenarios where one modality might be unavailable.


8.Inconsistent Claim of "Small-Scale Data": The paper repeatedly claims its method is designed for "small-scale data," which seems to contradict the actual dataset sizes. The experiments are conducted at the patch level, and according to Appendix H, the number of patches per dataset is substantial (71k, 126k, 40k). These figures do not align with a conventional definition of a small-scale dataset and could be misleading.


References:

[1] Xi, J., Osea, J., Xu, Z., & Hartford, J. S. (2024). Propensity score alignment of unpaired multimodal data.


[2] Hayat, N., Geras, K. J., & Shamout, F. E. (2022, December). MedFuse: Multi-modal fusion with clinical time-series data and chest X-ray images.

[3] Yao, W., Yin, K., Cheung, W. K., Liu, J., & Qin, J. (2024, March). Drfuse: Learning disentangled representation for clinical multi-modal fusion with missing modality and modal inconsistency.

**Questions:**

1.In Section 2.3, line 152, the phrase "they are unsuitable for patch-level tasks" is ambiguous. What does "they" refer to—the slide-level task or the MIL algorithms? Could you also clarify why it is necessary to specifically emphasize that slide-level analyses rely on MIL in this context?


2.The paper's core idea is to leverage unpaired data, yet it seems only the cycle consistency loss can operate on such data. The adversarial loss, as formulated, appears not to explicitly designed to utilize unpaired samples for alignment. Could you please clarify how the adversarial learning component directly benefits from the availability of unpaired data in your design?


3.Regarding the alignment loss (L_align): The formulation seems to omit the projection layers E_G and E_I, directly comparing the outputs of the shared encoder E_sh. Additionally, could you justify the choice of L2 loss for alignment over cosine similarity, which is a common and often more effective for aligning representations in a shared latent space?


4.Regarding Table 2, could you clarify the definition of "pairing percentage"? Does a 40% value mean that 40% of the training data is paired, with the remainder being unpaired? Furthermore, how were the baselines (CLIP and Concat) trained in these reduced pairing scenarios—were the unpaired samples simply discarded, or were they utilized in some other way?

---

### Official Review · Reviewer_KzpP · 2025-10-30

**Soundness:** 2
**Presentation:** 2
**Contribution:** 1
**Rating:** 2
**Confidence:** 4

**Summary:**

The paper proposes a multimodal approach that leverages adversarial learning to integrate patch-level histopathology and spatial transcriptomics. More specifically, the proposed CAMEO builds cross-modal GANs that learn shared and aligned representations from pre-trained unimodal encoders (vision and gene expression). The authors evaluate their approach across three datasets (lung, breast, and thymus), addressing niche classification and cell-type composition prediction. The reported results indicate that CAMEO can lead to well-aligned representations, albeit inconclusive wrt the choice of uni-modal encoders and/or approach to constructing the shared embedding.

**Strengths:**

- The motivation is clear; developing multimodal methods that can leverage abundant unimodal data with weakly paired samples addresses an important challenge in computational pathology.
- The modular design of CAMEO allows the use of different encoder combinations.

**Weaknesses:**

W1. The work follows a long line of research on combining whole slide imagery (slide-level and patch-level) and omic data. Some of these work like [1] have been mentioned; many recent methods (to name a few [2-8]) haven’t. But neither work has been discussed in relation to CAMEO, and why wasn’t it considered as a benchmark? Works like [1,2,4,5] address similar tasks to CAMEO; the others are general enough to be applied in a similar setting. In fact, it is unclear why CAMEO wasn’t evaluated on more “standard” benchmarks (in addition to the ones presented in the paper); as far as I understand, it should be readily applicable and would have better highlighted the role of the adversarial approach.

W2. Some design decisions seem very arbitrary and not well supported. For instance, there is a section on GAT, which seems like a main part of CAMEO; when in fact it is only one variant of the method, which is not even fully evaluated (only in 2/3 of the tasks). The authors make some claims and recommendations based on inconclusive findings, rather than providing further evidence and *actual* ablation studies of the method —for instance, the effects of different adversarial loss components, the size of the shared representations, or the probe vs. fine-tuned head…

W3. The CAMEO design does not seem scalable beyond two modalities. It seems that will require many more intermediate models (G()/D()) to capture and combine the different pair-wise modality combinations.


[1] Xie et al. “Spatially resolved gene expression prediction from HE histology
images via bi-modal contrastive learning” 2023

[2] He et al. “Integrating spatial gene expression and breast tumour morphology
via deep learning” 2020

[3] Hemker et al “HEALNet: Multi-modal Fusion for Heterogeneous Biomedical Data” 2024

[4] Nonchev et al. “DeepSpot: Leveraging Spatial Context for Enhanced Spatial
Transcriptomics Prediction from H&E Images” 2025

[5] Chung et al. “Accurate Spatial Gene Expression Prediction by Integrating Multi-Resolution Features” 2024

[6] Zhu et al. “Diffusion Generative Modeling for Spatially Resolved Gene Expression Inference from Histology Images” 2025.

[7] Chen et al. " Pan-cancer integrative histology-genomic analysis via multimodal deep learning” 2022

[8] Chen et al “ Multimodal Co-Attention Transformer for Survival Prediction in Gigapixel Whole Slide Images”

**Questions:**

- See weaknesses
- How sensitive is the multimodal alignment to the choice of unimodal encoders' embedding dimensions? Do mismatched dimensions affect performance?
- In many cases, at full pairing, concatenation outperforms/is comparable to CAMEO. This suggests that adversarial alignment may actually hurt performance when sufficient paired data is available. This needs some further clarification. Furthermore, the experiments with reduced pairing show an odd trend: some performance *improvements* with more unpaired data. This also needs further discussion and clarification.
- What does “patient-wise” stratification mean in this setting for train/val/test splitting?

---

### Official Review · Reviewer_Tqwx · 2025-10-31

**Soundness:** 3
**Presentation:** 3
**Contribution:** 2
**Rating:** 4
**Confidence:** 5

**Summary:**

This paper introduces CAMEO, a two-stage framework that (i) adversarially aligns frozen image and gene-expression encoders into a shared latent space using a WGAN-style objective plus cycle-consistency and a small paired-alignment loss, and then (ii) trains a linear head for niche-type classification and cell-composition regression. Experiments span three tissues (lung, breast, thymus) with two spatial-omics platforms (Xenium/Visium). The authors report that multimodal models generally beat unimodal ones; Concat is often the strongest at 100% pairing, but degrades as paired data decreases, while CAMEO is more robust under limited pairing (e.g., +4.4% Macro-F1 at 40% pairing for lung vs. −31.5% CLIP, −2.2% Concat; Table 2/Fig. 3). They also release a paired histology–transcriptomics dataset at the “niche” (224×224) level.

**Strengths:**

* The paper targets the common case where spatial-omics pairs are scarce and expensive, and it releases paired, annotated “niche” datasets (Xenium/Visium), which is valuable for reproducibility and downstream benchmarking.
* Three tissues, two tasks (niche classification, composition regression), and many encoder combinations under varying pairing ratios provide a wide lens on when multimodal helps and how robustness changes as pairing drops.

**Weaknesses:**

* **Adversarial alignment stability and identifiability remain under-analyzed.** GAN training is known to be unstable; WGAN/Gulrajani improves but does not eliminate failure modes, and adversarial objectives match distributions, not instance-level correspondences. The paper does not ablate λ-weights, discriminator/generator capacities, or show convergence/stability diagnostics; nor does it probe non-identifiability (e.g., ALICE-style joint matching issues) in this biomedical setting. [1-3]
* No visualizations of where cancer/niches are localized on slides or how gene pathways correspond spatially after alignment; comparable works commonly include qualitative maps or spatial correlation checks (e.g., HE2RNA’s virtual spatialization). Adding cross-modal nearest-neighbor retrievals, Grad-CAM/attention overlays, or gene-set heatmaps would strengthen claims.
* Beyond aggregate metrics, there is limited biological analysis (e.g., niche-specific marker recovery, pathway enrichment consistency). Methods like GLUE/SCOT emphasize structure-aware or OT-based correspondences that can be probed biologically; a similar probe here would help. [4,5]
* Section 4.5 varies encoder pairs, but there’s no toggle of the three losses (adversarial vs. cycle vs. paired-alignment), no swap of WGAN-GP vs. vanilla GAN, and no report of failure rates/mode collapse. This makes it hard to attribute robustness to the adversarial component rather than to frozen backbones or the small paired-alignment term.

[1] Zhu, J-Y. et al. “Unpaired Image-to-Image Translation using Cycle-Consistent Adversarial Networks (CycleGAN).” ICCV 2017.
[2] Arjovsky, M. et al. “Wasserstein GAN.” ICML 2017.
[3] Gulrajani, I. et al. “Improved Training of Wasserstein GANs (WGAN-GP).” NeurIPS 2017.
[4] Cao, Z-J. & Gao, G. “GLUE: Graph-linked unified embedding for integrating unpaired single-cell multi-omics.” Nat. Biotechnol. 2022.
[5] Demetci, P. et al. “SCOT: Single-Cell Multi-Omics Alignment with Optimal Transport.” J. Comput. Biol. 2022.

**Minor**

* HBM2 vs. HBM3 for 94 GB H100 (should be HBM3).

**Questions:**

* What drives robustness—adversarial, cycle, or paired-align loss? Add an ablation toggling each loss and a WGAN-GP variant; include retrieval metrics (image→genes, genes→image) to isolate instance-level alignment.
* How stable is training? Please report variance over runs, discriminator/generator losses, gradient-penalty settings, and any early-stopping heuristics. Can you include a failure-rate estimate (e.g., % runs that diverge)?

---

### Official Review · Reviewer_68Kn · 2025-11-01

**Soundness:** 2
**Presentation:** 2
**Contribution:** 2
**Rating:** 4
**Confidence:** 4

**Summary:**

this work tackles multimodal learning (2 modalities) with data that is partially paired.
authors propose an adversarial modality fusion framework to mitigate the missing pairing.
in particular, the fusion is done via 2 cross-modal generators + discriminators trained via standard adversarial and cycle consistency losses in addition to alignment loss; followed by a linear probing.
results are reported image histology and gene expression modalities using different backbones. ablations are also provided.

**Strengths:**

- the writing is good.
- the paper tackles an important issue which is learning with partially labeled data, especially in medical domain.
- results are reported in addition to ablations.
- release of dataset with images and gene expression.

**Weaknesses:**

- lack of justification. the method - sec.3 - is poorly justified. authors simply provided a series of steps without really justifying it. the same thing in introduction. the adversarial approach proposed here is not well aligned with existing works; and it seems to come from nowhere. authors simply said it is better in performance compared to contrastive method. why adversarial approach remains unclear.

- no comparison to existing works for multimodal learning with partially paired data. it is not clear how this method compares to previous works/SOTA.

- results are not consistent and sometimes strange. in table.1, with 100% paired data, the proposed method does not outperform others except in 2 cases. in tab.2, the proposed method yields better performance with having low paired data compared to full in the case of lung. something similar also in breast (60 vs 40 %).

**Questions:**

- please improve the justification of the method. and compare to SOTA and previous works.

**Details Of Ethics Concerns:**

none.

---

### Note · Authors · 2025-11-22

**Comment:**

We appreciate the time and constructive feedback provided by the reviewers. After careful consideration, we have decided to withdraw the submission. We believe that additional work is needed to integrate the suggested improvements and to strengthen the contribution. We plan to address these points and resubmit a revised version to a suitable venue.

**Withdrawal Confirmation:**

I have read and agree with the venue's withdrawal policy on behalf of myself and my co-authors.